# B6 Mouse Strain: The Best Fit for LPS-Induced Interstitial Cystitis Model

**DOI:** 10.3390/ijms222112053

**Published:** 2021-11-08

**Authors:** Ching-Hao Chen, Chun-Hou Liao, Kuo-Chiang Chen, Kuan-Lin Wang, Xiao-Wen Tseng, Wei-Kung Tsai, Han-Sun Chiang, Yi-No Wu

**Affiliations:** 1Department of Biomedical Science, Sheffield University, Sheffield S10 2TN, UK; tspc12345@gmail.com; 2Division of Urology, Department of Surgery, Cardinal Tien Hospital, New Taipei City 231, Taiwan; liaoch22@gmail.com; 3School of Medicine, Fu Jen Catholic University, New Taipei City 242, Taiwan; 062934@mail.fju.edu.tw (K.-C.C.); michael614188@gmail.com (K.-L.W.); 4Department of Urology, Cathay General Hospital, Taipei City 106, Taiwan; 053824@mail.fju.edu.tw; 5Program in Pharmaceutical Biotechnology, College of Medicine, Fu Jen Catholic University, New Taipei City 242, Taiwan; alitsen@yahoo.com.tw; 6Department of Urology, Mackay Memorial Hospital, Taipei City 104, Taiwan; weiko11@gmail.com; 7Ph.D. Program in Nutrition and Food Science, Graduate Institute of Biomedical and Pharmaceutical Science, Fu Jen Catholic University, New Taipei City 242, Taiwan; 8Department of Medicine, Mackay Medical College, New Taipei City 251, Taiwan; 9Mackay Junior College of Medicine, Nursing, and Management, Taipei City 104, Taiwan; 10Graduate Institute of Biomedical and Pharmaceutical Science, Fu Jen Catholic University, New Taipei City 242, Taiwan; 11Department of Urology, Fu Jen Catholic University Hospital, New Taipei City 242, Taiwan

**Keywords:** interstitial cystitis, lipopolysaccharide, FVB mouse strain, B6 mouse strain, cytometry

## Abstract

Interstitial cystitis (IC) is a chronic inflammatory disease characterized by bladder pain and increased urinary frequency. Although the C57BL/6J (B6) and FVB/NJ (FVB) mouse strains are commonly used as animal models for studies involving the urinary system, few reports have compared their lower urinary tract anatomy, despite the importance of such data. Our study aimed to characterize bladder function changes in FVB and B6 mouse strains with lipopolysaccharide (LPS)-induced IC, to understand mouse model-based bladder research. The bladder function parameters were measured by cystometrogram. Histological assay was examined by hematoxylin and eosin stain, Masson’s trichrome stain, and immunofluorescence staining. Results indicated that the two strains in the control group exhibited different bladder structures and functions, with significant anatomical differences, including a larger bladder size in the FVB than in the B6 strain. Furthermore, cystometry tests revealed differences in bladder function pressure. LPS-treated B6 mice presented significant changes in peak pressure, with decreased intercontraction intervals; these results were similar to symptoms of IC in humans. Each strain displayed distinct characteristics, emphasizing the care required in choosing the appropriate strain for bladder-model studies. The results suggested that the B6 mouse strain is more suitable for IC models.

## 1. Introduction

Interstitial cystitis (IC) is a chronic inflammation of the bladder, defined as the absence of infection or other determinable causes, accompanied by lower urinary tract symptoms for more than six weeks. The main symptoms of interstitial cystitis include chronic pelvic pain and urinary system symptoms, such as urgency or frequent urination. The three most common causes of this symptom are bladder urothelial dysfunction (bladder urothelial dysfunction), neurogenic inflammation, and neuropathic pain [1,2,3,4,5]. It stems from the inflammation and thinning of the protective glycosaminoglycan layer and causes heightened nerve sensitivity [6], amplified signals to the sixth level of the lumbar area nerve, and increased micturition frequency. Furthermore, the damaged glycosaminoglycan layer may become more permeable to urinary toxins, allowing the absorption of such toxins by the muscular layer, subsequently leading to loss of bladder function [5,7,8]. However, IC mechanisms are still theoretical with many unknowns, and researchers need to continue to explore treatment options, primarily with the help of animal models.

Lipopolysaccharide (LPS) causes inflammation of the detrusor smooth muscle and is commonly used to induce IC in animal subjects to promote the progress of tissue fibrosis [9,10,11]. Researchers have multiple model choices at their disposal for reproducing diseases in animals. However, the mouse model presents several advantages, including clear genetic pedigree, reproductive ease, minimal housing space, and significant genomic and anatomical similarities to humans, all of which help to reduce overall experimental costs [11,12]. In additional, knockout mice are the best research model for exploring the role of gene defects in IC.

C57BL/6J (B6) and FVB/NJ (FVB) are the most commonly used mouse strains for studying gene deficiency and the urinary system, but only a few studies have reported on the differences in their lower urinary tract anatomy and voiding function. A cystometrogram (CMG, cystometry test), which is commonly used to test bladder function, record bladder pressure, intercontraction intervals (ICIs), and voided volume per micturition [13,14], can be used to detect anatomical and functional differences between the bladders of the two strains. B6 mice are the most commonly used strain for cystometry studies. Meanwhile, FVB mice are used in the knockout model for mating and are generally not subjected to CMG testing. Previous reports stated different mouse strains had different drug sensitivities and pathological processes [15,16]. These results summarized that different mouse strains can perform different experimental functions; therefore, investigating and characterizing differences in bladder size, morphology, and function between B6 and FVB mice is essential for studies involving the urinary tract.

Our study aimed to explore the differences in normal bladder anatomy and in urination between FVB and B6 mouse strains. We also examined the change of histological characteristics and pathological processes after LPS induction. To the best of our knowledge, we are the first to characterize the bladder difference of the FVB and B6 mouse strains and to identify significant urination differences before and after IC induction. This study will supply basic urination data on the B6 strain animal model for the study of IC. Choosing an appropriate animal research model helps not only to better understand the pathological mechanism of IC, but it can also increase the possibility of clinical application.

## 2. Results

### 2.1. Gross Examination

As revealed by the gross examination shown in Figure 1, using an iron ruler for scale, the bladder of an FVB mouse is larger than that of the B6 mouse. Although a visual inspection cannot detect the relative volume of urine between the two bladders, a gross observation does reveal that the B6 strain bladder exhibits a thicker wall, as indicated by the FVB bladder’s greater transparency. In addition, a greater volume of urine may be implied from the FVB mouse’s larger bladder size. Tere were no notable diferences in the body appearance and weight between FVB and B6 mouse; however, it was observed that LPS induced mice were weighed signifcantly less than control mice (Appendix A).

### 2.2. CMG Testing of Control Group FVB and B6 Mice

CMG test results revealed distinguishing features between the FVB (FVB-ctrl) and B6 (B6-ctrl) mouse strains in the control group (Figure 2). Basal pressure, which is the baseline pressure of the bladder containing less urine, and non-voiding contraction (NVC) frequency of FVB-ctrl mice (Figure 2A) were, respectively, more stable and lower than those of B6-ctrl mice (Figure 2B). Statistical analysis results indicated that the basal pressure of the FVB-ctrl mice peaked higher at 8.4 cm H_2_O than that of the B6-ctrl mice, which peaked at 4.5 cm H_2_O, with the threshold pressure being approximately 17.2 and 8.6 cm H_2_O, respectively; here, threshold pressure was the pressure detected at the start of muscle contraction. However, the B6-ctrl mice presented a higher peak pressure (33.4 cm H_2_O) than that of the FVB-ctrl mice (29.7 cm H_2_O), with the FVB-ctrl mice requiring less pressure to contract the bladder, based on the delta pressure; here, peak pressure is the pressure required for urine output. Micturition frequency was higher in FVB-ctrl mice (Figure 2A) than in B6-ctrl mice (Figure 2B), with approximately three and four occurrences every ten minutes, respectively. ICI periods, which were 180 s in FVB-ctrl mice and 250 s in B6-ctrl mice, indicated that the B6 mice bladders filled more slowly.

### 2.3. LPS Treatment Effect on Bladder Function

Figure 3 compares CMG-assessed bladder functions between the LPS-treated and control group mice in each strain. Pressure peaks of LPS-treated FVB (FVB-LPS) and B6 (B6-LPS) mice are shown in Figure 3C,D, respectively. As compared to the corresponding control mouse group, these peaks are irregular with decreased ICI, indicating symptoms similar to those of IC (Figure 3A,B, respectively). Although IC was successfully induced in both mouse strains, CMG test results revealed significant differences. Basal pressure was not significantly impacted by LPS treatment in either strain and was measured at 8.27 and 3.97 cm H_2_O in FVB-LPS and B6-LPS mice, respectively. However, peak and threshold pressures significantly decreased with LPS treatment, suggesting LPS-induced bladder damage in both strains. B6-LPS mice exhibited greater decreases in threshold pressure and peak pressure, causing these pressures to become almost half of those exhibited by FVB-LPS mice. The ICI of both mouse strains decreased with LPS treatment compared with the corresponding control mouse group, with B6 mice exhibiting a greater decrease than that demonstrated by FVB mice. These CMG results indicate the B6 mouse strain is more sensitive to LPS than the FVB mouse strain.

### 2.4. Histology of Mouse Bladders

To evaluate the inflammation status after LPS administration on three principal tissue layers of the bladder wall in both mouse strains, bladder tissues from each mouse were collected for hematoxylin and eosin stain (H&E) staining. The presence of bladder inflammation and abnormal expression on bladder morphology after LPS induction were found in the B6 mouse strain (Figure 4). Figure 5 displays the Masson’s trichome staining of the bladders from both the control group and LPS-treated FVB and B6 mice, revealing various dissimilarities. The FVB-ctrl bladder exhibits more interstitial folds and a thicker interstitium and transitional epithelium than that in the B6-ctrl bladder. Masson’s trichome staining shows the thinner muscle layers of the FVB-ctrl bladder than that of the B6-ctrl bladder, with multiple junctions between muscle bundles. The FVB-LPS bladder exhibits a thinner interstitium with decreased collagen precipitation than that of the FVB-ctrl bladder. In both strains treated with LPS, the violet staining indicates spaces where muscle junctions have been replaced by collagen. Damage to the transitional epithelium layer by LPS treatment, which was revealed by the loss of its row-like structure into separate pieces, can be observed in both strains.

### 2.5. α-SMA Staining of Mouse Bladders

Figure 6 demonstrates the α-smooth muscle actin (α-SMA) immunostaining of the bladders from the four different groups—B6-ctrl, B6-LPS, FVB-ctrl, and FVB-LPS. The first column (B6-ctrl) and third column (FVB-ctrl) reveal several dissimilarities, as observed in the Masson’s staining. At 100× magnification, we observed that the muscular layer of the B6-ctrl bladder was thicker than that of the FVB-ctrl bladder. However, the junction structure did not correspond to the structure observed upon Masson’s staining. The histological section of the FVB-ctrl bladder in Figure 4 showed several gap junctions. However, the SMA staining showed fewer gap junctions in the FVB-ctrl bladder than in the B6-ctrl bladder. Further, the B6-ctrl bladder exhibited thicker muscle fibers than the FVB-ctrl bladder. After LPS treatment, the gap junctions in B6-LPS and FVB-LPS groups increased compared with the gap junctions in their respective controls, as shown in Figure 6.

### 2.6. Connexin Staining of Mouse Bladders

Figure 7 displays connexin, which detects gap junctions, in the bladders of B6-ctrl, B6-LPS, FVB-ctrl, and FVB-LPS mice. The connexin signal in the B6-ctrl bladder was stronger than that in the FVB-ctrl bladder, and the signal was detected around the muscle cells. This result corresponds to the SMA staining in Figure 6, which also shows that the number of gap junctions were fewer in the FVB-ctrl bladder than in the B6-ctrl bladder. In the LPS-treatment groups, the FVB-LPS bladder showed a stronger connexin signal than that observed in the FVB-ctrl bladder; however, the signal in the B6-LPS bladder was decreased than B6-ctrl group.

### 2.7. Staining of Tight Junction Protein in Mouse Bladders

Figure 8 shows the bladder sections stained with ZO-1, which stains the tight junction proteins between the bladder muscles. Similar to the connexin staining, ZO-1 signals were stronger in the B6-ctrl bladder than in the FVB-ctrl bladder, indicating more tight junctions in the bladder muscle of the B6-ctrl mouse than that of the FVB-ctrl mouse. In the B6 mice, there was not much variation in the ZO-1 signal between the LPS-treated and control groups. However, the ZO-1 protein signal was higher in sections from the FVB-LPS than that of the FVB-ctrl group.

### 2.8. Staining of Cytoplasmic Intermediate Filament Proteins in Mouse Bladders

Figure 9 shows keratin-5 (K5) staining to visualize cytoplasmic intermediate filament proteins on the bladder epithelial tissue in all the mouse groups. The B6-ctrl bladder section showed a stronger K5 signal, and thus a thicker epithelial tissue, than that of the FVB-ctrl. The keratin presented in the bladder wall epithelium is thought to protect the bladder. Contrary to previous knowledge, the FVB-LPS bladder showed a stronger K5 signal than the FVB-ctrl bladder. However, K5 expression was significantly decreased in LPS-treated B6 mice compared to the control B6 mice because some parts of urothelium had been destroyed by the LPS.

## 3. Discussion

In this study, we compared two mouse strains, FVB and B6, and determined that the strains exhibited completely dissimilar bladder functions and histological structures. The results of this research showed that the bladder structures of different mouse strains are different. It also showed a significant difference in bladder voiding patterns. After LPS induction, B6 strain mice showed symptoms that were more similar to clinical IC symptoms. In addition, H&E staining in B6 mice showed severe urothelial damage and leukocyte infiltration. We had verified the important molecular performance of the bladder tissue; B6 mice showed more significant expression than that of FVB before and after LPS induction. Although we successfully induced IC in both mouse strains, the impact on B6 mice was greater, with more apparent ICI changes. The decrease in ICI leads to increased urinary frequency, which is the primary symptom of IC.

Studies have revealed that bladder size does not vary significantly between human races [3,12]. However, our gross examinations of the FVB and B6 mouse strains revealed a notable difference in bladder size. Most studies have used the B6 mouse strain rather than the FVB strain for CMG modeling but lacked evidence regarding any advantages of using the B6 strain over the FVB strain. Our study reports the differences in bladder structure and functions between the two strains since such information is important in assessing the most appropriate mouse strain for research [17].

The present study demonstrated that the bladder of the FVB mouse strain is larger in volume than that of the B6 mouse strain; however, the latter exhibits longer ICI times, indicating the ability to retain urine for longer periods of time. Multiplying the CMG saline input rate, which was approximately 30 μL/min, by the ICI time gives the volume of urine stored in the bladder. The ICI of the FVB-ctrl and B6-ctrl mice, represented by the gap between the urine output peaks, was approximately 180 s and 250 s, respectively, equating to bladder volumes of 90 μL and 125 μL, respectively. However, the values of ICI and volume did not correspond to the gross and histological findings, demonstrating that different mouse strains effectively exhibit different functions. To better assess which mouse strain would be the most appropriate model choice for IC studies, we induced IC in FVB and B6 mouse models via LPS treatment for two weeks and assessed the scale of the bladder damage by observing the changes in bladder pressure and histology. When subjected to LPS-induced IC, both mouse strains exhibited decreases in ICI and the corresponding pressure [4,5]. The bladder function loss caused by LPS-induced IC can be observed in Figure 3C,D. The FVB-LPS ICI data, as displayed in the chart, does not demonstrate a clear decrease; however, the NVC increased within the gaps between the peaks. B6-LPS mice exhibited increased irregularity in pressure peaks, also accompanied by increased NVC, decreased peak pressure to approximately half that of untreated B6 mice, and decreased ICI [10]. In comparing the impact of LPS-induced IC on bladder pressure, the FVB CMG results did not change sharply with LPS treatment. Although results did differ, the subtle differences were not easy to detect. However, B6 CMG data changed dramatically as a result of LPS treatment, especially ICI decrease, which corresponds to the primary IC symptom of increased urinary frequency [7].

CMG results demonstrated that different mouse strains exhibit different bladder functions, and that LPS successfully induced IC in both mouse strains. We then observed and compared the histology results of control and experimental FVB and B6 mouse groups. In the LPS-treated group, the epithelium was damaged, which may have led to urine absorption by the interstitium and muscle layer. The bladder interstitium contains an abundance of nerves, which may be stimulated by urine toxin units to signal bladder muscle contraction for urine voiding. In addition, the toxin units can lead to muscle damage and inflammation [18]. The present study successfully induced IC in both mouse strains, as indicated by the observed decrease in ICI, leading to increased urinary frequency, bladder inflammation, and possibly, discomfort. The loss of bladder function is similar to the symptoms of IC in humans [5,19]. Increased collagen in the interstitium and muscle layer indicates interstitium and muscle layer fibrosis and loss of bladder contraction ability [20].

To observe the abundance of collagen and cell junctions between the bladder’s muscle cells, we stained the bladder sections with connexin or ZO-1 [21,22]. We observed a greater decrease in connexin and ZO-1 signal in the bladder tissue of B6-LPS than in the B6-ctrl, which indicates that B6-LPS over-activity caused by the LPS treatment. In contrast, both the connexin and ZO-1 signals were higher in the bladder muscle tissue of FVB mice treated with LPS than that of the FVB-ctrl. Hence, these results show that inducing IC by LPS treatment is more efficient in the B6 strain than the FVB strain since IC progresses faster in the B6 than in the FVB mice strain. The histology data of the tissue sections were in agreement with the ICI data, which showed that after treatment with LPS, the ICI of B6-LPS decreased sharply. As connective tissues such as the gap and tight junctions increases and replaces the normal parenchymal tissue, it may cause considerable tissue remodeling and lead to the formation of a permanent scar tissue, if left unchecked [18,23]. Therefore, it seems that the fibrosis process in the B6 mice bladder upon LPS treatment was faster than in the FVB-LPS bladder, as shown by the decrease in connexin and ZO-1 signal, as well as of the ICI value. However, the FVB-LPS mouse bladder tissues were still undergoing fibrosis, since the connexin and ZO-1 signals were higher in the FVB-LPS bladder compared to that in the FVB-ctrl. Therefore, the B6 strain is a better choice as a model for studying IC than the FVB strain.

However, there are some limitations in this study. We compared only two different mouse strains; BALB/c mice are not appropriate for IC induction because of their different immunologic reactivity, especially when challenged with LSP. They display a different polarization that means they are mainly used in experimental models for Th2-related disorders, such as asthma [24] or food allergy [25].

The histology results of our study were unable to determine the type of increased collagen. Atallah et al. demonstrated that varying circumstances produce different types of organ collagen; however, they could not identify which collagen variety leads to fibrosis [18]. Future research may focus on identifying the most prevalent type of collagen present in IC-damaged layers [19,26]. In additional, we only evaluated effect of LPS induction in this study; more data will need to be determined in future studies.

In summary, our study revealed the different bladder structures and functions in two mouse strains, which will enable researchers to identify the most appropriate strain for bladder-model studies. We also concluded that the B6 mouse strain is more suitable for IC model studies.

## 4. Materials and Methods

### 4.1. Animal Models

All mice were purchased from BioLASCO Taiwan Co., Ltd. (Taipei, Taiwan), and approved by the Fu Jen Catholic University Animal Care and Use Committee (IACUC approval No. A10726. The approval date was on 28 August 2018). All animal experiment methods were conducted in accordance with the guidelines and regulations established by Fu Jen Catholic University, Taiwan.

### 4.2. Study Design

A total of 18 mice were randomly divided into two groups. The control group, which did not undergo LPS treatment, contained four mice of each strain, B6 and FVB. The experimental group, which was be treated with LPS, contained five B6-strain mice and five FVB-train mice. All mice were subjected to CMG and histological analysis.

### 4.3. Mouse Surgery and CMG

Before surgery, each mouse was anesthetized with Zoletil-50 (tiletamine/zolazepam) (1 mL/kg) via intraperitoneal injection. Mouse abdomen hair was shaved to perform an incision at the body midline, and the bladder was carefully pushed out of the body for photography. Using forceps to pinch the top of the bladder dome, 6-0 nonabsorbable polypropylene sutures (Prolene, Cornelia, GA, USA) were passed through the bladder. An incision was made near the suture for placement of a polyethylene micro-tubing 50 catheter into the bladder, the end of which was heated to form a small cuff. The catheter and bladder were bound with the sutures, the muscle layer closed, followed by combination of the catheter with a polyethylene micro-tubing 90 to close the skin layer.

After surgery, each mouse was placed in the CMG testing cage and allowed to rest for approximately 1 h. Upon regaining consciousness, the mouse was placed on the injector machine for CMG parameter detection by the MP3 pressure transducer (Biopac Systems Inc., Santa Barbara, CA, USA), matched with the Biopac Student lab 4.1 (Biopac Systems Inc., Santa Barbara, CA, USA) for data capture. The CMG parameters assessed included basal pressure, peak pressure, threshold pressure, and ICI.

### 4.4. Mouse IC Induced by the LPS

LPS from *Escherichia coli* (Sigma-Aldrich, St. Louis, MO, USA) was mixed with phosphate-buffered saline (PBS), at a concentration of 50 µg/mL. Dosages of 0.5 mg/kg were administered to each mouse in the experimental group for two weeks. After LPS treatment, bladder function was once again assessed via CMG.

### 4.5. Histology Examination

After CMG testing, the mice were euthanized via administration of high-dose pentobarbital sodium solution, and their bladders were collected and fixed in 10% formaldehyde (*w*/*v*) for 24 h. The bladders were then cut in half, dehydrated, post-fixated, and embedded in paraffin blocks. For embedment, a paraffin block was placed in a stainless-steel box on a heat plate at approximately 50–60 °C. A bladder specimen was placed into the center of the liquid wax and covered by a plastic embed box. After the wax was allowed to cool, the embed box was covered and placed into the refrigerator for 15 min. Once the wax had solidified, excess wax was cut from the embed box. The embedded tissue was cut into 5 µm sections, adhered to charged slides, and dewaxed with xylene. The hydrated bladder tissue was processed in graded alcohol (100%, 95%, 80%, and 70%) and ddH_2_O for 5 min, followed by H&E and Masson’s trichrome staining.

### 4.6. Immunofluorescence Staining

Mouse bladder tissues were immersed in optimal cutting temperature (OCT) (Thermo Fisher Scientific, Waltham, MA, USA) compound and frozen immediately using liquid nitrogen. The frozen tissue blocks were sectioned using a cryostat into 3 µm slices. The tissue sections were placed in a heater for 1 h, followed by three PBS washes for 10 min each. The sections were then covered by a solution containing 0% goat serum/2% bovine serum albumin/0.2% Triton X-100 (Sigma-Aldrich) and incubated at room temperature for 1 h. Then, the tissue sections were incubated overnight at 4 °C with any of the four different primary antibodies, depending on the experiment—rabbit anti-α-smooth muscle actin (SMA, ab5694; Abcam, Cambridge, UK), rabbit anti-Connexin-43 (ab11370; Abcam, Cambridge, UK), rabbit anti-ZO-1 (40-2200; Thermo Fisher Scientific, Waltham, MA, USA), and rabbit anti-Keratin-5 (KRT-5, TA300887; OriGene, Rockville, MD, USA). After staining with the primary antibody, the tissue sections were incubated in secondary antibody for 1 h at room temperature. For sections stained by anti-SMA primary antibody, a 1:500 dilution of Alexa Fluor 594 secondary antibody was used, while for the others, the Alexa Fluor 488 secondary antibody was used. The sections were imaged at 100× magnification using a fluorescence microscope. All the figures were analyzed and merged using ImageJ software (National Institutes of Health, Bethesda, MD, USA).

## Figures and Tables

**Figure 1 ijms-22-12053-f001:**
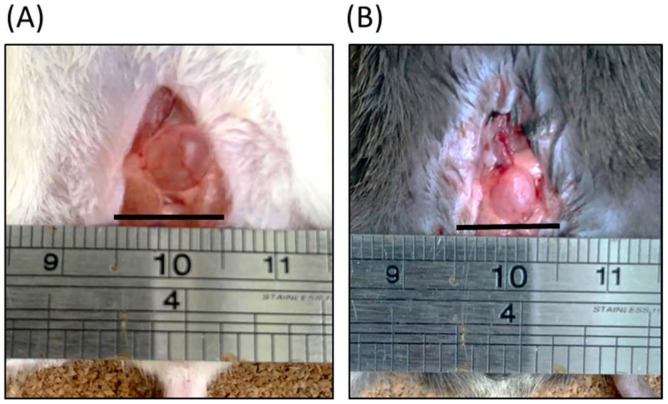
Gross examinations of bladders of (**A**) FVB/NJ (FVB, white) and (**B**) C57BL/6J (B6, black) mice. The iron ruler is displayed for scale. FVB bladder was larger than B6 bladder. Black bar indicates one centimeter.

**Figure 2 ijms-22-12053-f002:**
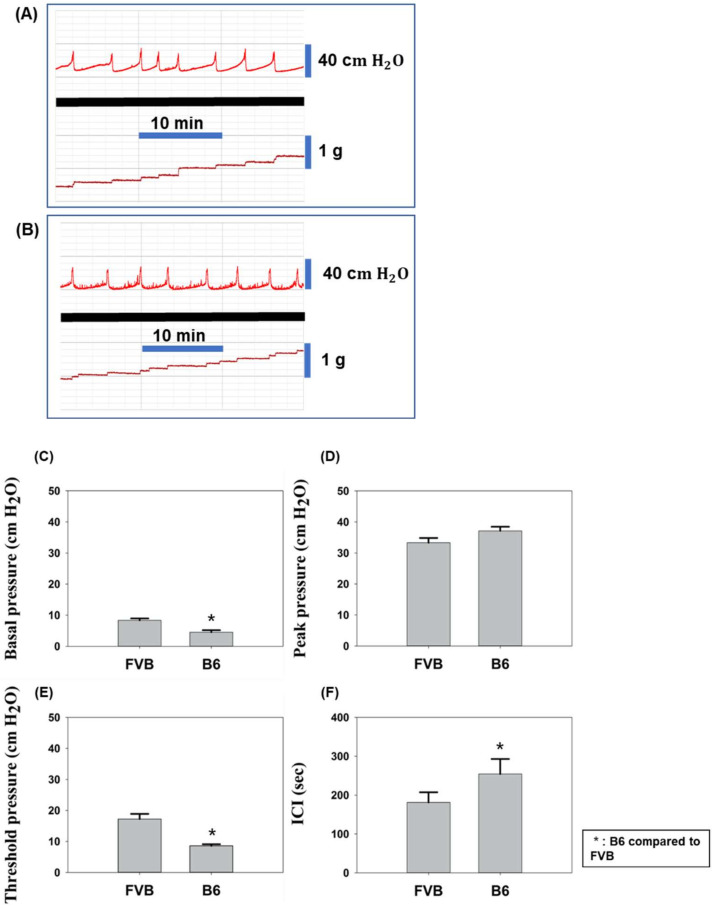
Cystometrogram (CMG) data line charts of (**A**) FVB and (**B**) B6 mice from control groups. Bladder pressure is shown above the bold black line, and output volume below. CMG test results of both strains are shown for (**C**) basal pressure, (**D**) peak pressure, (**E**) threshold pressure, and (**F**) intercontraction interval (ICI).

**Figure 3 ijms-22-12053-f003:**
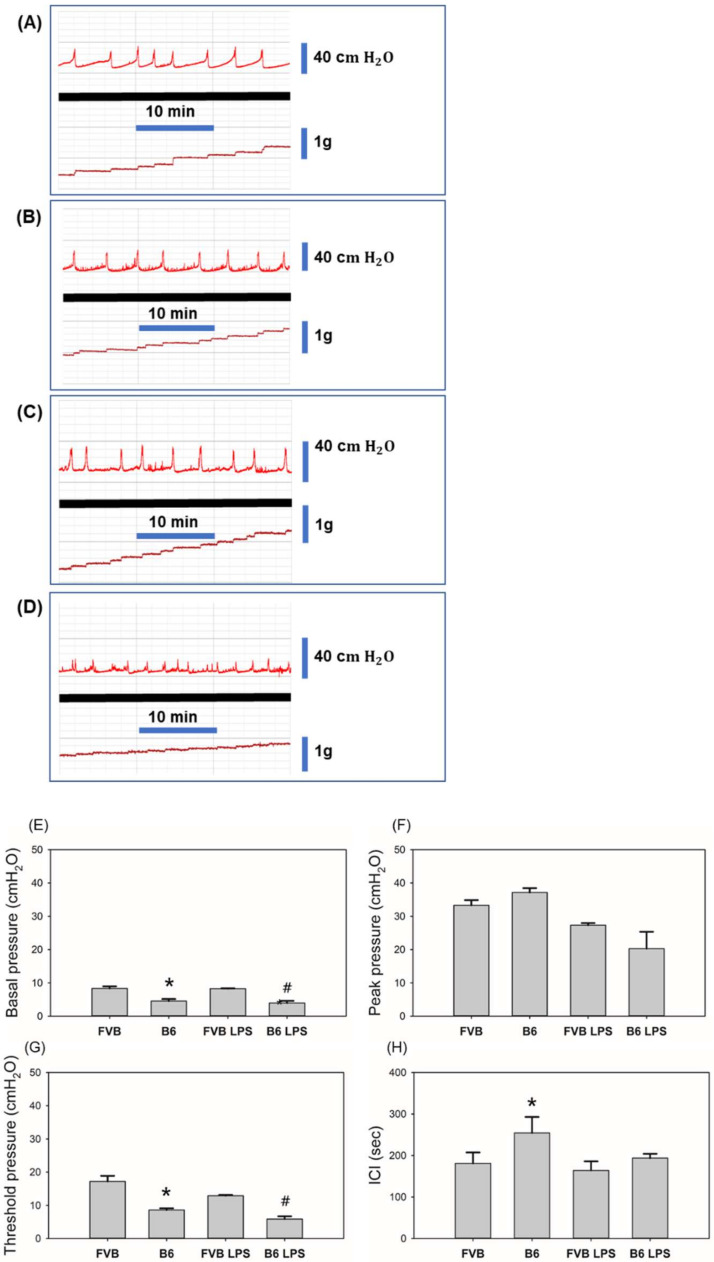
CMG data line charts of (**A**) FVB control, (**B**) B6 control, (**C**) lipopolysaccharide (LPS)-treated FVB, and (**D**) LPS-treated B6 mice. Bladder pressure is shown above the bold black line, and output volume below. CMG test results of all groups are shown for (**E**) basal pressure, (**F**) peak pressure, (**G**) threshold pressure, and (**H**) ICI. * B6 compared to FVB; # B6 LPS compared to FVB LPS.

**Figure 4 ijms-22-12053-f004:**
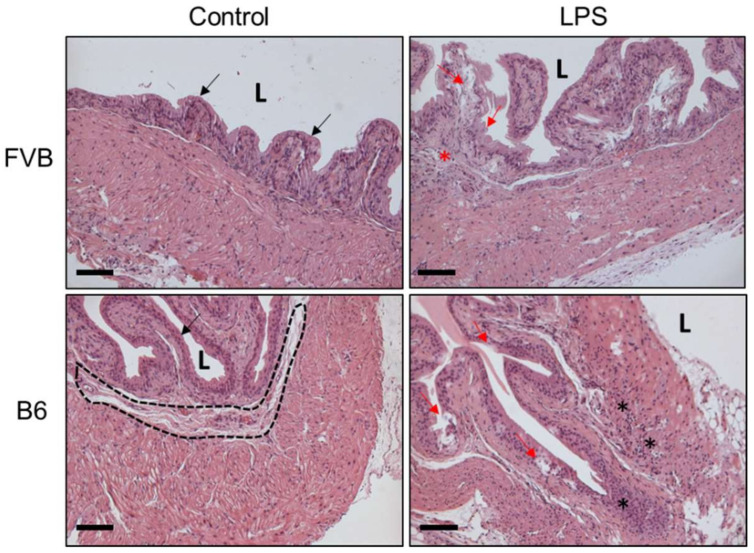
Bladder H&E staining of control group and LPS-induced group of both FVB and B6 mice. Black arrow indicates the urothelium of bladder. Black dotted circle shows the interstitial layer. Red arrow indicates the damaged urothelium of bladder after LPS-induced IC. Black star shows the leukocyte infiltration site. Red star shows the edema site. Lumen = L. Scale bar of 100 μm.

**Figure 5 ijms-22-12053-f005:**
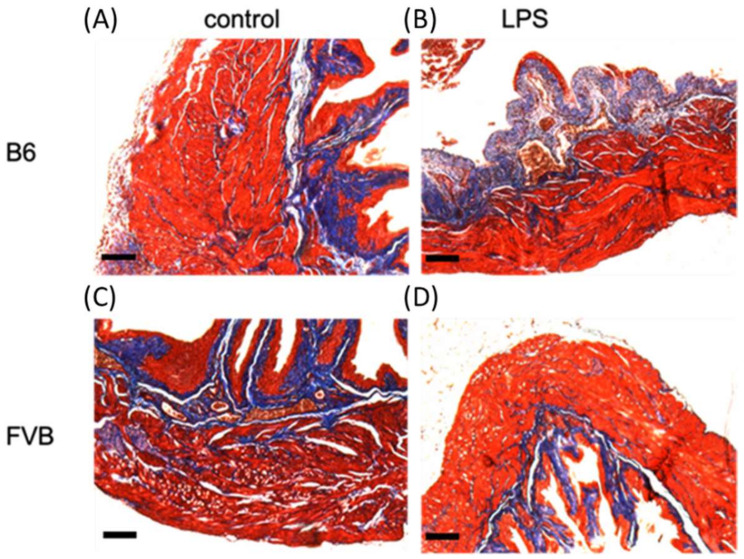
Bladder histology of (**A**) B6-ctrl, (**B**) B6-LPS mice, (**C**) FVB-ctrl and (**D**) FVB-LPS. The FVB mouse bladder exhibits a larger lumen layer than the B6 mouse bladder. Under magnification, the differences between the muscular layer of each strain bladder becomes clear; the B6 bladder exhibits a tighter muscular layer and the FVB bladder exhibits a muscular layer with multiple gaps between each muscle cell.

**Figure 6 ijms-22-12053-f006:**
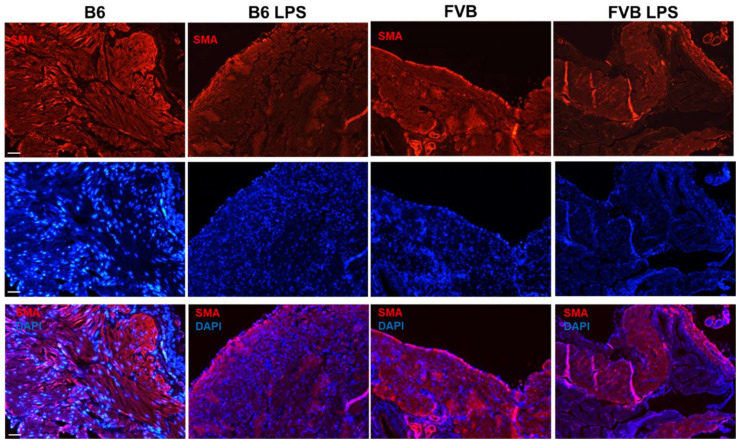
Immunofluorescent staining of α-smooth muscle actin (SMA; red, first row) in bladders from the B6-ctrl, B6-LPS, FVB-ctrl, and FVB-LPS mice. DAPI staining (blue) of the samples is shown in the second row, while the merged images of SMA and DAPI are shown in the third row for each group. All images were acquired at 100× magnification (Scale bar = 50 µm).

**Figure 7 ijms-22-12053-f007:**
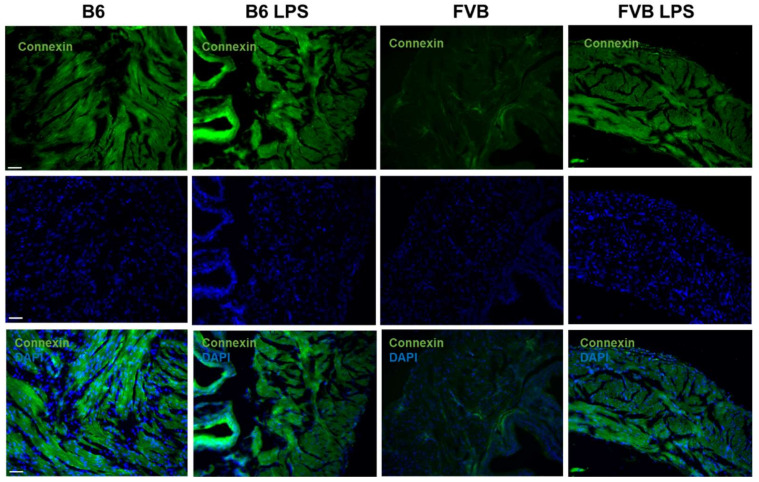
Connexin staining (green, first row) showing gap junctions in the frozen sections from B6-ctrl, B6-LPS, FVB-ctrl, and FVB-LPS mice’s bladders in the row. DAPI staining and the connexin–DAPI merged images for each group are shown in the second and third row, respectively. All images were acquired at 100× magnification (Scale bar = 50 µm).

**Figure 8 ijms-22-12053-f008:**
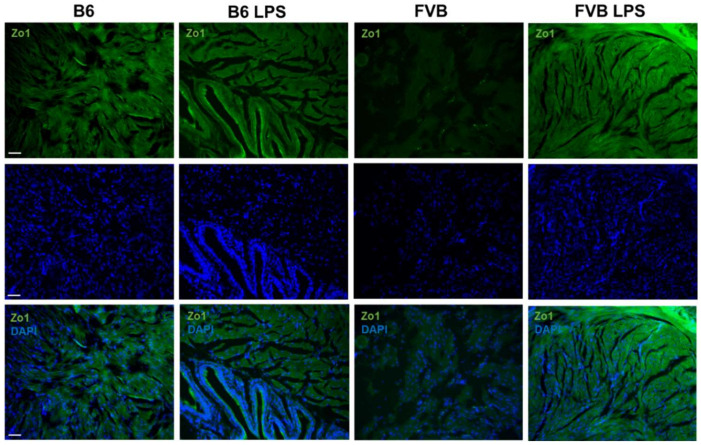
Immunostaining of zonula occludens-1 (ZO-1, green, first row) for the visualization of tight junction proteins in the bladders of B6-ctrl, B6-LPS, FVB-ctrl, and FVB-LPS mice. DAPI staining and the ZO-1 and DAPI merged images for each group are shown in the second and third row, respectively. All images were acquired at 100× magnification (Scale bar = 50 µm).

**Figure 9 ijms-22-12053-f009:**
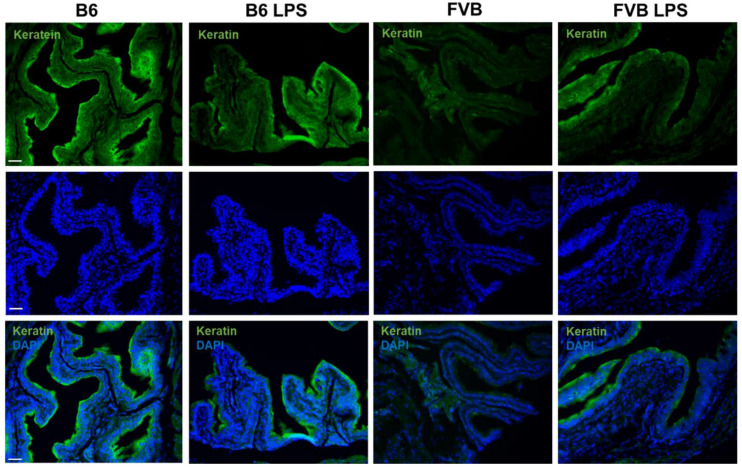
Keratin-5-stained cytoplasmic intermediate filament proteins (green, first row) expressed on the bladder epithelial tissues of the B6-ctrl, B6-LPS, FVB-ctrl, and FVB-LPS mice. DAPI-stained and the keratin 5-DAPI merged images for each group are shown in the second and third row, respectively. All images were taken at 100× magnification (Scale bar = 50 µm).

## Data Availability

The data presented in this study are available on request from the corresponding author.

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
