# Peer review of "B6 Mouse Strain: The Best Fit for LPS-Induced Interstitial Cystitis Model"

_ijms, 2021, doi:10.3390/ijms222112053_

Round 1
Reviewer 1 Report
- In Figure 1. If size bar inserted (or with ruler) in the photos (instead of cotton swab), itis helpful to readers.
- If IC induced by LPS differently affects to the physiological and/or biochemical status of mice, time course of the weight changes of mice should be recorded.
- If possible, the changes of size of bladder according to the inflammation level should be measured in the animal model.
- The expression level of inflammatory cytokines such as TNF-alpha and interlukin-10 should be checked in LPS-induced IC.
Author Response
Reviewer 1:
1. In Figure 1. If size bar inserted (or with ruler) in the photos (instead of cotton swab), itis helpful to readers.
Ans: Thanks for your suggestion. We had renewed the figure 1 and put the correct scale bar in the photo.
2. If IC induced by LPS differently affects to the physiological and/or biochemical status of mice, time course of the weight changes of mice should be recorded.
Ans: Thank you for your suggestion. We have added the body weight record in supplementary figure 1.
3. If possible, the changes of size of bladder according to the inflammation level should be measured in the animal model.
Ans: Thank you for your comment. Our research in this article focused on the difference between the normal bladder volume and tissue between B6 and FVB strain mouse, and evaluated which mouse model is the best choice for IC symptoms after LPS induction. Therefore, this study did not thoroughly explore the inflammation performance and influence of different mouse strains. In this study, we added the HE staining to evaluate the expression of leukocyte infiltration. We will evaluate the differences in bladder inflammation induced by LPS in different strains of mice in future studies.
4. The expression level of inflammatory cytokines such as TNF-alpha and interlukin-10 should be checked in LPS-induced IC.
Ans: Thanks for your recommendations. In this study, we are going to focus on the IC symptoms in different mouse strain which is best fit for IC and which can be the best model for following research. Selecting the best fit model is important for the LUTS research and the clinical application. Those markers are quite important for discovering the inflammation in IC, we will evaluate these marker differences in different strains of mice in future studies.

Reviewer 2 Report
Minor revisions
-line 30 “cystometry”
-line 83: error at the beginning of the sentence
-why the authors decided to examine 15 mice? What about the sample size calculation?
-I suggest to move “However, the histology results of our study were unable to determine the type of 334 increased collagen. Atallah et al demonstrated that varying circumstances produce differ-335 ent types of organ collagen; therefore, it cannot identify which collagen variety increases 336 in the layer, leading to fibrosis [5]. Future research may focus on identifying the most 337 prevalent type of collagen present in IC-damaged layers [26,27].” form conclusion section to discussion section. In my opinion, in the conclusion section authors should focus on the last comment about the quality of the manuscript, the most relevant results and the further perspectives.
-there are some grammar and lessical errors that I suggest to correct
Major revisions
-In the discussion section I suggest to compare data directly and with specific correlations to the other experiences in literature and with numerical data rather than only “speculative” comparisons.
-Moreover, authors compared only two different mice strain. In my opinion, the title should reflect this consideration and deserves to be changed
Author Response
Reviewer 2
Minor revisions
1. line 30 “cystometry”
Ans: Thanks for your comment. Those vocabulary error has corrected.
2. line 83: error at the beginning of the sentence
Ans: Thanks for your comment. Those vocabulary error has corrected.
3. why the authors decided to examine 15 mice? What about the sample size calculation?
Ans: Thank for the comment. In this experiment, there were 4 mice in the group of control of B6 and FVB, respectively. After that, 5 B6 mice and 5 FVB mice were induced IC. So, the correct sample calculation is 18 and we had corrected it. This study explored the differences of the mouse strains on development of disease model that can reference gene deficiency experiment (n>3). In additional, the n value (4-5) of each group had reached the statistical difference in the study.
4. I suggest to move “However, the histology results of our study were unable to determine the type of 334 increased collagen. Atallah et al demonstrated that varying circumstances produce differ-335 ent types of organ collagen; therefore, it cannot identify which collagen variety increases 336 in the layer, leading to fibrosis [5]. Future research may focus on identifying the most 337 prevalent type of collagen present in IC-damaged layers [26,27].” form conclusion section to discussion section. In my opinion, in the conclusion section authors should focus on the last comment about the quality of the manuscript, the most relevant results and the further perspectives.
Ans: thanks for your comment. We also notice that our conclusion is really too long and its make sense to move “However, the histology results of our study were unable to determine the type of increased collagen. Atallah et al demonstrated that varying circumstances produce different types of organ collagen; therefore, it cannot identify which collagen variety increases in the layer, leading to fibrosis [5]. Future research may focus on identifying the most prevalent type of collagen present in IC-damaged layers [26,27].” to the discussion. As you said, moving those words to discussion can let the reader more focus on the difference bladder size may affect the function in bladder.
5. There are some grammar and lessical errors that I suggest to correct.
Ans: Thanks for your comment. Those grammars and vocabulary errors had corrected and proof-reading by the native speaker.
Major revisions
1. In the discussion section I suggest to compare data directly and with specific correlations to the other experiences in literature and with numerical data rather than only “speculative” comparisons.
Ans: Thanks for your comment. The discussion section had been revised as your suggestion.
2. Moreover, authors compared only two different mice strain. In my opinion, the title should reflect this consideration and deserves to be changed
Ans: Thanks for your comment. These two mouse strains are mostly used in genetically deficient mice, so we compared the differences between these two mouse strains. The title just only emphasized that the B6 strain is suitable for the establishment of IC model and evaluation of pathogenesis.

Reviewer 3 Report
- In the introduction and results (see the first sentence here, for instance) there are several grammar inconsistencies and/or missing words. The English writing should be carefully revised.
- I think that authors should revise the introduction in order to create the appropriate and clear background for the appropriate understanding of this study. Indeed, it sounds a little dispersive at the moment and some important information are missing. For instance, the authors should clarify what is the clinical relevance of IC, in order to emphasize the translational value of their research. (e.g. Int J Urol. 2019 Jun;26 Suppl 1:17-19. doi: 10.1111/iju.13969; Semin Reprod Med. 2018 Mar;36(2):123-135. doi: 10.1055/s-0038-1676089)
- I think that the long second paragraph should be summarized, and some points may be resumed in the discussion more in detail.
- I recommend putting the materials and methods before the results. It does not make any sense to put them between results and discussion.
- In the results, can the authors also describe the simple histology with EoH staining, in order to understand the usual inflammatory infiltration after LPS treatment? Can the authors describe the composition and location of the inflammatory infiltrate, if any, and include some images? This aspect in not clear and it is actually important for a readership that is not very focused on this topic.
- In the discussion, the authors could briefly mention and explain why other mice strains ae not used or are not suitable for studies on IC. For instance, explain why BALB/c mice are not appropriate. Is that because of their different immunologic reactivity, especially when challenged with LSP? Indeed, they display a different polarization that make them mainly used in experimental models for Th2-related disorders, like asthma (refer: J Biol Regul Homeost Agents. 2018 Mar-Apr;32(2):335-339. PMID: 29685015) or food allergy (see: J Dairy Sci. 2021 Sep 15:S0022-0302(21)00881-X. doi: 10.3168/jds.2021-20260.). Please, briefly explain this important point in the discussion.
- As said, some concepts emerged in the introduction could be more suitable to the discussion.
- I would suggest the authors to clearly state and list their main findings in the beginning of the introduction and, then, discuss them one by one methodically. Therefore, the discussion should be reorganized and expanded.
- There is no mention to any limitations or gap related to this study. Are there any? If no, please state that. If yes, please list and briefly described them.
- The conclusion is too long and should include any references, which may be used in the discussion. Here the authors are supposed to provide only the final and clear conclusions and take-home messages. Indeed, the second paragraph seems to mention a limitation of this study or something that the authors have not addressed in the present study.
- Funding: the authors here state: “This research received no external funding”, which is not believable. Indeed, in the “Acknowledgments” the authors declare a grant support: “This study was supported by grants from the Cardinal Tien Hospital 354 (CTH106A-2A27)”. This grant must be reported in the funding statement, of course.
- please, also report the date of the IACUC approval, in addition to the number.
- references: to be completed and updated according to the reorganization of the manuscript and the previous comments and suggestions.
Author Response
Revierwe 3
1. In the introduction and results (see the first sentence here, for instance) there are several grammar inconsistencies and/or missing words. The English writing should be carefully revised.
Ans: Thanks for your comment. The grammar and the missing words in the introduction and results have been proof-reading by the native speaker.
2. I think that authors should revise the introduction in order to create the appropriate and clear background for the appropriate understanding of this study. Indeed, it sounds a little dispersive at the moment and some important information are missing. For instance, the authors should clarify what is the clinical relevance of IC, in order to emphasize the translational value of their research. (e.g. Int J Urol. 2019 Jun;26 Suppl 1:17-19. doi: 10.1111/iju.13969; Semin Reprod Med. 2018 Mar;36(2):123-135. doi: 10.1055/s-0038-1676089)
Ans: Thank for your comment. We had revised the introduction section to emphasize the translational value of their research as your suggestion.
3. I think that the long second paragraph should be summarized, and some points may be resumed in the discussion more in detail.
Ans: Thank for your comment. We had revised the introduction section as your suggestion.
4. I recommend putting the materials and methods before the results. It does not make any sense to put them between results and discussion.
Ans: Thanks for your comment. Putted the materials and methods section after the result is the request in ijms journal manuscript format.
5. In the results, can the authors also describe the simple histology with EoH staining, in order to understand the usual inflammatory infiltration after LPS treatment? Can the authors describe the composition and location of the inflammatory infiltrate, if any, and include some images? This aspect in not clear and it is actually important for a readership that is not very focused on this topic.
Ans: Our research in this article focused on the difference between the normal bladder volume and tissue between B6 and FVB strain mouse, and evaluated which mouse model is the best choice for IC symptoms after LPS induction. Therefore, this study did not thoroughly explore the inflammation performance and influence of different mouse strains. In this study, we added the HE staining to evaluate the expression of leukocyte infiltration. We will evaluate the differences in bladder inflammation induced by LPS in different strains of mice in future studies.
6. In the discussion, the authors could briefly mention and explain why other mice strains ae not used or are not suitable for studies on IC. For instance, explain why BALB/c mice are not appropriate. Is that because of their different immunologic reactivity, especially when challenged with LSP? Indeed, they display a different polarization that make them mainly used in experimental models for Th2-related disorders, like asthma (refer: J Biol Regul Homeost Agents. 2018 Mar-Apr;32(2):335-339. PMID: 29685015) or food allergy (see: J Dairy Sci. 2021 Sep 15:S0022-0302(21)00881-X. doi: 10.3168/jds.2021-20260.). Please, briefly explain this important point in the discussion.
Ans: Thank you for your kindly suggestion. We had explained the reason about BALB/c mice are not appropriate for the establishment of IC model in the discussion section.
7. As said, some concepts emerged in the introduction could be more suitable to the discussion
Ans: Thanks for your comment again. We had revised the introduction and summarized the introduction to make the structure clearly. Also, some points which contain in the introduction will put into discussion because they may be more suitable.
8. I would suggest the authors to clearly state and list their main findings in the beginning of the introduction and, then, discuss them one by one methodically. Therefore, the discussion should be reorganized and expanded.
Ans: Thanks for your comment. We had rearranged the structure of the discussion and make the statement clearly, easy to be following, also list the main results in the first paraphrase.
9. There is no mention to any limitations or gap related to this study. Are there any? If no, please state that. If yes, please list and briefly described them.
Ans: Thanks for your comment. We had listed the limitations of this study in the discussion.
10. The conclusion is too long and should include any references, which may be used in the discussion. Here the authors are supposed to provide only the final and clear conclusions and take-home messages. Indeed, the second paragraph seems to mention a limitation of this study or something that the authors have not addressed in the present study.
Ans: Thanks for your comment. The conclusion section had been revised.
11. Funding: the authors here state: “This research received no external funding”, which is not believable. Indeed, in the “Acknowledgments” the authors declare a grant support: “This study was supported by grants from the Cardinal Tien Hospital 354 (CTH106A-2A27)”. This grant must be reported in the funding statement, of course.
Ans: Thanks for your comment. The funding section had been revised.
12. please, also report the date of the IACUC approval, in addition to the number.
Ans: Thanks for your comment. The date of the IACUC approval had added in the materials and methods.
13. references: to be completed and updated according to the reorganization of the manuscript and the previous comments and suggestions.
Ans: Thanks for your comment. We had reorganized the references section.
Round 2
Reviewer 3 Report
I do not have additional major comments.